# MarginGAN: Adversarial Training in Semi-Supervised Learning

**Jinhao Dong**
School of Computer Science and Technology,
Xidian University
Xi'an 710126, China
jhdong@stu.xidian.edu.cn

**Tong Lin**[*]
Key Laboratory of Machine Perception, MOE
School of EECS, Peking University, Beijing,
& Peng Cheng Laboratory, Shenzhen
lintong@pku.edu.cn

## Abstract

A Margin Generative Adversarial Network (MarginGAN) is proposed for semi-supervised learning problems. Like Triple-GAN, the proposed MarginGAN consists of three components—a generator, a discriminator and a classifier, among which two forms of adversarial training arise. The discriminator is trained as usual to distinguish real examples from fake examples produced by the generator. The new feature is that the classifier attempts to increase the margin of real examples and to decrease the margin of fake examples. On the contrary, the purpose of the generator is yielding realistic and large-margin examples in order to fool the discriminator and the classifier simultaneously. Pseudo labels are used for generated and unlabeled examples in training. Our method is motivated by the success of large-margin classifiers and the recent viewpoint that good semi-supervised learning requires a "bad" GAN. Experiments on benchmark datasets testify that MarginGAN is orthogonal to several state-of-the-art methods, offering improved error rates and shorter training time as well.

## 1   Introduction

In the real world, unlabeled data can usually be obtained relatively easily, while manually labeled data costs a lot. Therefore, **semi-supervised learning** (SSL), which learns by using large amounts of unlabeled data with limited labeled data, meets a variety of practical needs.

Pseudo labels are artificial labels of unlabeled data to play the same role as labels of manually annotated data, which is a simple and effective method in semi-supervised learning. Several traditional SSL methods, such as self-training [1–3] and co-training [4], are based on pseudo labels. In the past few years, deep neural network has made a great advancement in SSL, and hence the idea of pseudo labels is incorporated into deep learning to leverage unlabeled data. In [5] the class with the maximum predicted probability is picked as the pseudo label. Temporal Ensembling proposed in [6] uses a ensemble prediction as the pseudo label, which is an exponential moving average of label predictions on different epochs, under different regularization and input augmentation conditions. In contrast to [6] where label predictions are averaged, in the *Mean Teacher* approach [7] model weights are averaged instead. The role pseudo labels play in [5] and [6, 7] is not exactly the same. Pseudo labels in [5] have the identical effect with ground-truth labels to minimize the cross-entropy loss, whereas pseudo labels in [6, 7] serve as targets for the prediction to achieve consistency regularization, which can make the classifier give consistent outputs for similar data points.

Recently, generative adversarial networks (GANs) has been applied to SSL and obtained amazing results. The method of *Feature Matching* (FM) GANs proposed in [8] substitutes the original binary

---

[*]T. Lin is the corresponding author.

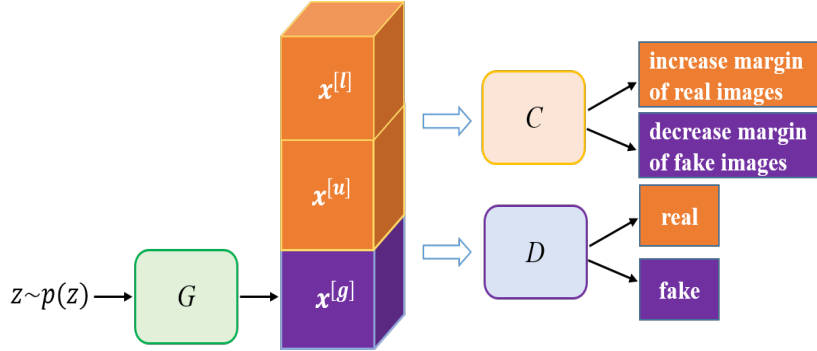

Figure 1: The architecture of MarginGAN.

discriminator with a $(K+1)$-class classifier. The aim of the classifier (i.e. the discriminator) is to classify labeled samples into the correct class, unlabeled samples into any of the first K classes and generated samples into the $(K+1)$-th class. As a improvement of feature matching GANs, the method proposed in [9] verifies that good semi-supervised learning requires a "bad" generator. The proposed complement generator can yield artificial data points in low-density areas, thus encouraging the classifier to place the class boundaries in these areas and to improve the generalization performance.

Although the idea of using pseudo labels is simple and effective in deep learning, sometimes it might happen that the incorrect pseudo labels will impair the generalization performance and slow down the training of deep networks. Prior works such as [6, 7] make efforts on how to improve the quality of pseudo labels. Inspired by [9], we propose a method that encourages the generator to yield "bad" examples in SSL, so as to increase the tolerance of incorrect pseudo labels and reduce the error rates further.

To address the issue caused by the incorrect pseudo labels, we present MarginGAN, a GAN model in semi-supervised learning based on margin theory of classifiers. MarginGAN consists of three components—a generator, a discriminator and a classifier (See Fig. 1 for the architecture of Margin-GAN). The role of the discriminator is same as in standard GAN, distinguishing whether a example is from the real distribution or produced by the generator. The multi-class classifier is trained to increase the classification margin of real examples (including labeled samples and unlabeled samples), and decreases the margin of generated fake examples meanwhile. The goal of the generator is to yield bogus examples that look like realistic and have large margin, aiming at deceiving both the discriminator and the classifier simultaneously.

The paper is organized as follows. Section 2 briefly reviews related work, and our proposed Margin-GAN is described in Section 3. Experimental results are presented in Section 4. Section 5 concludes this paper.

## 2   Related Work

The research of semi-supervised learning dates back to 1970s, and there have emerged many classical SSL algorithms since then. Self-training [1–3] is probably the first SSL algorithm, which selects the unlabeled examples with the surest predictions and puts them into the labeled sample set in each iteration. Co-training [4] trains two classifier on two different views of the labeled examples respectively, and each classifier puts the unlabeled examples with the highest prediction confidence into the labeled dataset of the other classifier. The above two methods can be regarded as methods using pseudo labels. Graph-based semi-supervised learning [10, 11] constructs a neighborhood graph according to the geometric structure between the samples, and propagates the label from labeled samples to unlabeled samples utilizing the adjacency relation on the graph based on manifold hypothesis.

Recently, deep neural networks have made a great progress in SSL. As already discussed, the idea of pseudo labels is also used in [5–7]. To reduce the instability brought by pseudo labels, a coefficient $\alpha(t)$ is used in [5] to balance labeled samples and unlabeled examples. $\alpha(t)$ is set slowly increased so that the low weights can reduce the negative effect of unreliable pseudo labels. In [6, 7] the quality

of pseudo labels is improved by keeping an exponential moving average of predictions or model weights, respectively. Our work focuses on another perspective that the classifier learns from auxiliary generated examples. Besides [9], virtual adversarial training proposed in [12, 13] is similar to our approach in spirit.

Recent methods leveraging GANs achieve amazing results in SSL. It is worth noting that CatGAN proposed in [14] shares a similar flavor to our margin-based method. The classifier of CatGAN minimizes the conditional entropy of real samples, while maximizing the conditional entropy of generated fake examples at the same time. Triple-GAN [15] also adopts a three-player architecture, where the generator and the classifier characterize the conditional distributions between examples and labels, and the discriminator solely focuses on identifying fake example-label pairs. Triangle-GAN [16] develops a more complex architecture consisting of two generators and two discriminators. In [17] Structured Generative Adversarial Networks (SGANs) are proposed for semi-supervised conditional generative modeling, which can better manipulate the semantics of generated examples.

Besides the methods mentioned above, there has been other efforts in semi-supervised learning using deep generative models. In [18] the Ladder network was extended to the area of SSL. In [19] an unsupervised regularization term was proposed to explicitly enforce that the predictions of the multi-class classifier should be mutually-exclusive.

# 3 The Proposed MarginGAN

## 3.1 Motivation and Intuition

In a usual GAN model, the goal is to train a generator that can produce realistic fake examples such that a discriminator can not discern real or fake examples. However in SSL problems our purpose is to train a high-accuracy classifier achieving large margins of training examples. We hope that the generator can yield "informative" examples near the true decision boundary, just like support vectors in the SVM models. Here another kind of adversarial training arises: the generator attempts to produce large-margin fake examples, while the classifier aims at achieving small-margin predictions over these fake examples.

Wrong pseudo labels of unlabeled examples (and fake examples) greatly deteriorate the accuracy of prior methods based on pseudo labels, but our MarginGAN exhibits a better tolerance to wrong pseudo labels. Since the discriminator plays the same role in a usual GAN, we argue that the improved accuracy obtained by MarginGAN comes from adversarial interactions between the generator and the classifier.

First, the extreme training case in our ablation study (in Sec. 4.2) show that fake examples generated by MarginGAN can aggressively remedy the influence of wrong pseudo labels. Because the classifier enforces the small margin values of fake examples, the generator must yield fake examples near the "correct" decision boundaries. This will refine and shrink the decision boundary for surrounding the real examples.

Second, we illustrate the large-margin intuition on a four-class problem in Fig. 2. If the classifier chooses to believe the wrong pseudo labels, the decision boundaries have to stride over the "real" gap between the two classes of examples. But wrong pseudo labels lead to reduced values in margin, which hurts the generalization accuracy. Therefore, large margin classifiers should ignore those wrong pseudo labels for achieving higher accuracy.

## 3.2 Margin

**Definition of margins**   In machine learning, the margin of a single data point is defined to be the distance from that data point to a decision boundary, which can be used to bound the generalization error of the classifier. Both support vector machines (SVM) and boosting can be explainable with margin-based generalization bounds.

In the AdaBoost algorithm, $h_t(x) \in \{1, -1\}$ is a base classifier acquired in iteration $t$ and $\alpha_t \geq 0$ is its corresponding weight assigned to $h_t$. The combined classifier $f$ is a weighted majority vote of $T$

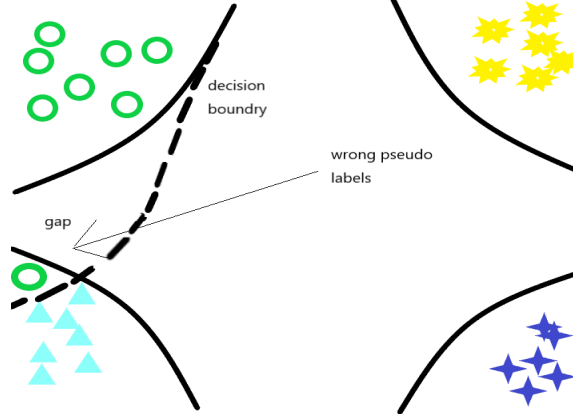

Figure 2: A illustration to show that wrong pseudo labels may cause decision boundaries across the true gap between two classes. A large-margin classifier would disregard those wrong pseudo labels to get better generalization.

base classifiers, which is formulated as

$$f(x) = \frac{\sum_{t=1}^{T} \alpha_t h_t(x)}{\sum_{s=1}^{T} \alpha_s}.$$

In [20] the margin of an instance-label pair $(x, y)$ is defined as

$$yf(x) = \frac{y \sum_{t=1}^{T} \alpha_t h_t(x)}{\sum_{s=1}^{T} \alpha_s}. \tag{1}$$

The sign of the margin reflects whether or not the prediction of the combined classifier is correct, while the magnitude indicates the prediction confidence. It is interesting that Eq. 1 can bring a unified form for both boosting and SVM:

$$\frac{y \langle \boldsymbol{\alpha}, \boldsymbol{h}(x) \rangle}{\|\boldsymbol{\alpha}\| \, \|\boldsymbol{h}(x)\|},$$

where $\boldsymbol{h}(x) \doteq [h_1(x), h_2(x), ..., h_T(x)]$ and $\boldsymbol{\alpha} \doteq [\alpha_1, \alpha_2, ..., \alpha_T]$. For boosting, $\ell_\infty$ norm is used for $\boldsymbol{h}$ and $\ell_1$ norm is used for $\boldsymbol{\alpha}$; and for SVM, $\ell_2$ norm is used for both $\boldsymbol{h}$ and $\boldsymbol{\alpha}$.

**Margins in semi-supervised learning** [21] propose the margin of an unlabeled example denoted as $|f(x)|$, that can be also represented as $\widetilde{y}f(x)$ with pseudo label $\widetilde{y} = \text{sign}(f(x))$. This way just regards the current prediction is correct and makes the classifier more certain of what it predicts currently. Regardless of labeled and unlabeled examples, larger margins of data points can decrease the upper bound of the generalization error, which brings about better generalization performance.

**Margins of multi-class classification** We set our problem in multi-class problems: for an instance-label pair $(x, y)$, $y \in \mathbb{R}^k$ is the ground-truth label in one-hot encoding and $C(x) \in \mathbb{R}^k$ is the prediction of the multi-class classifier $C$. The last layer of the classifier network is usually a softmax layer, so the output is a discrete distribution such that $\sum_{i=1}^{k} C_i(x) = 1$. The margin in multi-class problems is defined as the difference between the probability for the true class and maximal probability for the false classes:

$$\text{Margin}(x, y) = C_y(x) - \max_{i \neq y} C_i(x). \tag{2}$$

It is evident that if the margin is a large positive number, the probability of the correct class is peaked in the distribution $[C(x)_1, C(x)_2, \ldots, C(x)_k]$, indicating that the classifier is confident of its prediction. On the contrary, if the margin is a small positive number, the distribution is flat and the classifier is uncertain of its prediction, which has a similar flavor to CatGAN proposed in [14]. The classifier makes a mistake decision when the margin is negative.

### 3.3 Architecture Overview

The original architecture of GAN consists of two components, a generator and a discriminator, that play a zero-sum game. The generator $G$ transforms a latent variable $z \sim p(z)$ to a fake example $\hat{x} \sim p_g(\hat{x})$ such that the generated distribution $p_g(\hat{x})$ approximates the real data distribution $p(x)$. The discriminator $D$ is to distinguish generated fake examples from real examples. For adapting to semi-supervised learning, we add a classifier $C$ to the original architecture, which forms a three-player game. We retain the discriminator for encouraging the generator to produce visually realistic examples. We describe each of the components as follows. The architecture of MarginGAN is depicted in Fig. 1.

### 3.4 Discriminator

The discriminator $D$ of MarginGAN receives three kinds of inputs, labeled examples $x$ drawn from $p^{[l]}(x)$, unlabeled examples $\widetilde{x}$ drawn from $p^{[u]}(\widetilde{x})$ and generated examples $G(z) = \hat{x} \sim p_g(\hat{x})$. In our SSL setting, the discriminator should regard both labeled examples and unlabeled examples as real data points, and discern generated examples as fake data points. We define the loss function for the discriminator as

$$Loss(D) = -\big\{E_{x \sim p^{[l]}(x)}[\log(D(x))] + E_{\widetilde{x} \sim p^{[u]}(\widetilde{x})}[\log(D(\widetilde{x}))] + E_{z \sim p(z)}[\log(1 - D(G(z)))]\big\}.$$

### 3.5 Classifier

We add a multi-class classifier $C$ to the original GAN, as high-accuracy classification is our purpose in SSL. We develop the classifier from the perspective of margins. The classifier receives the same inputs as the discriminator—labeled examples, unlabeled examples and generated fake examples. In the following we will detail the corresponding objective of the classifier for each input.

For labeled examples, the classifier has the same objective as ordinary multi-class classifiers. Given an instance-label pair $(x, y)$, the classifier $C$ attempts to minimize the cross-entropy loss between the true label $y$ and the predicted label $C(x)$:

$$Loss_{CE}(y, C(x)) = -\sum_{i=1}^{T} y_i \log(C(x)_i),$$

where $y \in \mathbb{R}^k$ is in one-hot encoding, and $C(x) \in \mathbb{R}^k$ is the prediction. And the loss function for the labeled examples can be formulated as

$$\begin{aligned} Loss(C^{[l]}) &= E_{(x,y) \sim p^{[l]}(x,y)}\left[Loss_{CE}(y, C(x))\right] \\ &= E_{(x,y) \sim p^{[l]}(x,y)}\left[-\sum_{i=1}^{k} y_i \log\big(C(x)_i\big)\right]. \end{aligned} \quad (3)$$

Note that minimizing the cross-entropy encourages the increase of the probability of the true class and inhibits the probability of other false classes, leading to a larger margin defined in Eq. 2.

For unlabeled examples, the goal of the classifier is to increase the margin of these data points. However, since there is no information about the corresponding true label, we have no idea which class probability should be peaked. Like [5–7, 21], we leverage the pseudo label $\widetilde{y}^{[u]} \in \mathbb{R}^k$ in one-hot encoding for unlabeled examples. That is, the $\widetilde{y}^{[u]}$ vector has 1 at the only entry corresponding to the class with the maximum predicted probability of the current predict $C(\widetilde{x})$, while other entries are exactly zeros. With pseudo labels, we can increase the margin by minimizing the cross-entropy between $\widetilde{y}^{[u]}$ and $C(\widetilde{x})$. Intuitively, this objective will reinforce the confidence of the current predictions. The loss function is given as

$$\begin{aligned} Loss(C^{[u]}) &= E_{\widetilde{x} \sim p^{[u]}(\widetilde{x})}\left[Loss_{CE}(\widetilde{y}^{[u]}, C(\widetilde{x}))\right] \\ &= E_{\widetilde{x} \sim p^{[u]}(\widetilde{x})}\left[-\sum_{i=1}^{k} \widetilde{y}_i^{[u]} log(C(\widetilde{x})_i)\right], \end{aligned} \quad (4)$$

which has the same form as Eq. 3.

When it comes to generated examples, the classifier should decrease the margin of these data points and make the prediction distribution flat. The generated examples are another form of unlabeled data, and we take the same way to use pseudo label. In order to decrease the margin of generated examples, we introduce a new loss function, Inverted Cross Entropy (ICE) between two distributions

$$Loss_{ICE}(p, q) = -\sum_{i=1}^{k} p_i \log(1 - q_i),$$

where $p, q \in \mathbb{R}_k$. Minimizing the inverted cross entropy will increase the cross-entropy between the pseudo label $\widetilde{y}^{[g]}$ and $C(G(z))$, so that the prediction distribution will be flat and the margin will be decreased. The loss function for generated examples is

$$Loss(C^{[g]}) = E_{z \sim p(z)} \left[ Loss_{ICE}(\widetilde{y}^{[g]}, C(G(z))) \right]$$

$$= E_{z \sim p(z)} \left[ -\sum_{i=1}^{k} \widetilde{y}_i^{[g]} log(1 - C(G(z))_i) \right]. \tag{5}$$

Combining three loss functions defined in Eq. 3, 4 and 5 altogether, we obtain the integrated loss function of the classifier

$$Loss(C) = Loss(C^{[l]}) + Loss(C^{[u]}) + Loss(C^{[g]}). \tag{6}$$

## 3.6 Generator

The purpose of $G$ is to produce bogus examples that looks like realistic to the discriminator $D$ and improves the generalization of classifier $C$ meanwhile. From this perspective, $C$ and $D$ form an alliance to compete with $G$, while $G$ attempts to fool both $C$ and $D$. On the one hand, just like the standard GAN, the objective of $G$ is to generate fake data points that $D$ can not distinct. On the other hand, because $C$ increases the margin of real examples and decreases the margin of fake examples, $G$ should compete to yield data points having large margin to fool $C$. Therefore, in order to fool both $D$ and $C$, $G$ tries to yield realistic and large-margin examples simultaneously such that the generated fake data points can not easily separated from real examples. In a nutshell, the loss function of $G$ is formulated as

$$Loss(G) = -E_{z \sim p(z)} \left[ \log \left( D(G(z)) \right) \right] + E_{z \sim p(z)} \left[ Loss_{CE}(\widetilde{y}^{[g]}, C(G(z))) \right].$$

## 3.7 Minimax Game

To adapt the loss function of MarginGAN into a similar form of the original GAN, we combine all the loss functions of each component into a minimax problem:

$$\min_G \max_{D,C} J(G, D, C)$$

$$= \left\{ E_{x \sim p^{[l]}(x)}[\log(D(x))] + E_{\widetilde{x} \sim p^{[u]}(\widetilde{x})}[\log(D(\widetilde{x}))] + E_{z \sim p(z)}[\log(1 - D(G(z)))] \right\}$$

$$- \left\{ E_{(x,y) \sim p^{[l]}(x,y)}[Loss_{CE}(y, C(x))] + E_{\widetilde{x} \sim p^{[u]}(\widetilde{x})} \left[ Loss_{CE}(\widetilde{y}^{[u]}, C(\widetilde{x})) \right] \right.$$

$$\left. + E_{z \sim p(z)} \left[ Loss_{ICE}(\widetilde{y}^{[g]}, C(G(z))) \right] \right\},$$

where the first part is a minimax game between $G$ and $D$, and the second part is between $G$ and $C$. Instead, we can view the minimax game from the perspective of margin:

$$\min_G \max_{D,C} J(G, D, C)$$

$$= \left\{ E_{x \sim p^{[l]}(x)}[\log(D(x))] + E_{\widetilde{x} \sim p^{[u]}(\widetilde{x})}[\log(D(\widetilde{x}))] + E_{z \sim p(z)}[\log(1 - D(G(z)))] \right\}$$

$$+ \left\{ E_{(x,y) \sim p^{[l]}(x,y)} \left[ \text{Margin}(x, y) \right] + E_{\widetilde{x} \sim p^{[u]}(\widetilde{x})} \left[ \text{Margin}(\widetilde{x}, \widetilde{y}^{[u]}) \right] \right.$$

$$\left. + E_{z \sim p(z)} \left[ 1 - \text{Margin}(G(z), \widetilde{y}^{[g]}) \right] \right\},$$

if we redefine $\text{Margin}(x, y) \doteq \langle y, \log C(x) \rangle$ and $1 - \text{Margin}(x, y) \doteq \langle y, \log(1 - C(x)) \rangle$. In practice, each time any of three networks ($D$, $C$ and $G$) is trained with gradient descent over one example or a mini-batch with other two networks being fixed, same as the training procedure of an usual GAN.

Table 1: Error rates (%) on MNIST with 100, 600, 1000 and 3000 labeled examples in semi-supervised learning. The results of competing methods come from [5]. Means and stardard errors are reported for our method on 5 runs.

| # of labels | 100 | 600 | 1000 | 3000 |
|---|---|---|---|---|
| NN | 25.81 | 11.44 | 10.70 | 6.04 |
| SVM | 23.44 | 8.85 | 7.77 | 4.21 |
| CNN | 22.98 | 7.68 | 6.45 | 3.35 |
| TSVM | 16.81 | 6.16 | 5.38 | 3.45 |
| DBN-rNCA | — | 8.70 | — | 3.30 |
| EmbedNN | 16.86 | 5.97 | 5.73 | 3.59 |
| CAE | 13.47 | 6.30 | 4.77 | 3.22 |
| MTC | 12.03 | 5.13 | 3.64 | 2.57 |
| dropNN | 21.89 | 8.57 | 6.59 | 3.72 |
| +PL | 16.15 | 5.03 | 4.30 | 2.80 |
| +PL+DAE | 10.49 | 4.01 | 3.46 | 2.69 |
| MarginGAN (ours) | $3.53 \pm 0.57$ | $3.03 \pm 0.60$ | $2.87 \pm 0.71$ | $2.06 \pm 0.20$ |

## 4 Experiments

### 4.1 Preliminary Experiment on MNIST

Similar to our work, pseudo labels are employed in prior work [5] and experiments on MNIST are reported. To show the improvement brought by MarginGAN clearly, we first conduct a preliminary experiment on MNIST. We use the generator and the discriminator from the infoGAN [22], and use a simple convolutional network with six layers as the classifier. Although the classifier we use might be powerful than that used in [5], the subsequent ablation study can reveal the contribution brought by generated fake examples.

MNIST consists of a training set of 60,000 images and a test set of 10,000 images, with all images of 28 by 28 gray-scale pixels. In settings, we sample 100, 600, 1000 or 3000 labeled examples and use the rest of the training set as unlabeled samples. When training, we first pretrain the classifier to achieve the error rate lower than 8.0%, 9.3%, 9.5% and 9.7%, only with the labeled examples, respectively corresponding to 100, 600, 1000 and 3000 labeled examples. Then, the unlabeled samples and generated samples engage in the training process. Table 1 compares our results against other competing methods from [5]. We can see that the proposed MarginGAN outperforms these pseudo label based previous methods on each setting, which can be attributed to the participation of generated fake examples. Although this comparison with dated algorithms is somewhat unfair, our method does achieve higher accuracies under all settings and the subsequent ablation study further verifies the improvements of our method.

### 4.2 Ablation Study on MNIST

To find out the influence of labeled examples, unlabeled examples and generated fake examples, we ran ablation experiments with one or several types of examples fed as input at a time. In the ablation study, because of the instability of pseudo labels and lack of labeled examples in some cases, we decrease the learning rate from 0.1 to 0.01. We measured the lowest error rates and time consumed to training convergence in different settings, and the results are reported in Table 2.

**Unlabeled examples** Unlabeled examples plays an important role in semi-supervised learning. We can see that the addition of unlabeled examples can reduce the error rate from 8.21% to 4.54% with 3.67% improvement. To verify the uncertainty of the correctness of pseudo labels, we conducted an extreme attempt: the classifier was pretrained to achieve the error rate of 9.78% ($\pm$ 0.14%), and then we fed the classifier with unlabeled examples alone. In other words, the classifier can not access the labeled examples again. To our surprise, the error rate blew up and quickly reached to over 89.53%. The incorrect pseudo labels will mislead the classifier and hinder its generalization.

Table 2: Ablation study of our algorithm on MNIST. The amount of labeled examples in this experiment is 600. The abbreviations of L, U and G correspond to labeled, unlabeled and generated examples, respectively. The last two rows show an extreme training situation.

|  | Settings | Error Rates (%) | Training Time (sec.) |
|---|---|---|---|
| Normal Training | L | $8.21 \pm 0.82$ | $408.41 \pm 26.17$ |
|  | L + U | $4.54 \pm 0.41$ | $1305.64 \pm 495.18$ |
|  | L + U + G | $3.20 \pm 0.62$ | $367.79 \pm 82.82$ |
| Extreme Training | U | $89.53 \pm 0.81$ | — |
|  | U + G | $7.40 \pm 5.01$ | $886.83 \pm 193.98$ |

Table 3: Means and standard errors of the error rates (%) on SVHN and CIFAR-10 over 4 runs.

| METHODS | SVHN (500 labels) | CIFAR-10 (1000 labels) | CIFAR-10 (4000 labels) |
|---|---|---|---|
| Ladder [18] | — | — | $20.04 \pm 0.47$ |
| CatGAN [14] | — | — | $19.58 \pm 0.58$ |
| FM GANs [8] | — | — | $18.63 \pm 2.32$ |
| Triple-GAN [15] | — | — | $18.82 \pm 0.32$ |
| SGAN [17] | — | — | $17.26 \pm 0.69$ |
| $\prod$ model [7] | $6.83 \pm 0.66$ | $27.36 \pm 1.20$ | $13.20 \pm 0.27$ |
| MarginGAN (ours) | $6.07 \pm 0.43$ | $10.39 \pm 0.43$ | $6.44 \pm 0.10$ |

**Generated fake examples**   We fed generated examples to the classifier, making it robust to wrong pseudo labels and improving the performance. We can see that, compared with training of only labeled samples and unlabeled samples, the generated examples can further improve the error rates from 4.54% to 3.20%. Moreover, it's worth noting that the generated examples can remarkably reduce the training time consumed by 71.8%. However when we continue to train, the error rate starts to increase and overfitting arises. When the generated images are more realistic gradually, the classifier still reduces their margins, which might harm the performance. Back to the extreme situation mentioned above, when combining unlabeled images and generated images after pretraining, the error rates can be improved indeed (from 9.78% to 7.40%).

## 4.3   Results on SVHN and CIFAR-10

Next we run our MarginGAN method on two standard datasets in SSL—SVHN and CIFAR-10. We employ a 12-block residual network [23] with Shake-Shake regularization [24] as our classifier, which is same as the ResNet version used in the mean teacher implementation. Our algorithm integrates the generator and the discriminator from the infoGAN [22] into this residual network. We also use the mean teacher training for averaging model weights over recent training examples.

The details of SVHN and CIFAR-10 datasets are as follows. SVHN contains of 73,257 digits for training and 26,032 digits for testing, with each digit being a $32 \times 32$ RGB image. The CIFAR-10 dataset consists of 50,000 training samples and 10,000 test samples of $32 \times 32$ color images of 10 object classes. On SVHN we randomly select 500 labeled samples. And we use 1,000 and 4,000 labels to train on CIFAR-10, respectively. Table 3 shows the results of experiments on the SVHN and CIFAR-10 datasets. We can see the improvements brought by our method.

## 4.4   Generated Fake Images

We show the images generated by MarginGAN in Fig. 3 when the accuracy of classifier is increasing. As we can see, these fake images really look "bad": for instance, most generated digits in MNIST and SVHN are close to the decision boundaries such that one can not determine their labels with high confidence. This situation meets with our motivation of this paper.

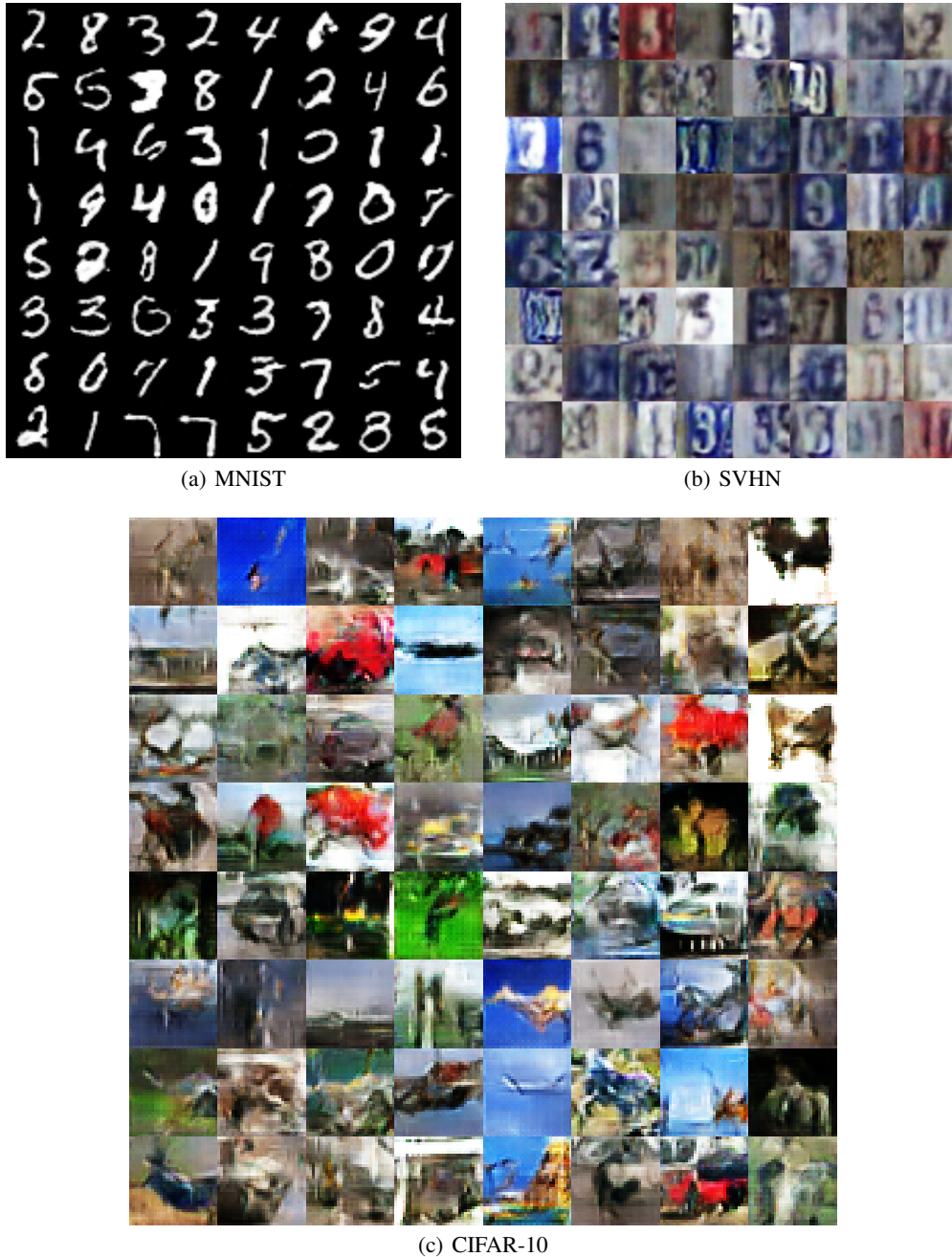

(a) MNIST

(b) SVHN

(c) CIFAR-10

Figure 3: Generated fake images by MarginGAN.

## 5 Conclusion

In this work, we presented the Margin Generative Adversarial Network (MarginGAN), which consists of three players—a generator, a discriminator and a classifier. The key is that the classifier can leverage fake examples produced by the generator to improve the generalization performance. Specifically, the classifier aims at maximizing margin values of true examples and minimizing margin values of fake examples. The generator attempts to yield realistic and large-margin examples to fool both the discriminator and the classifier. The experimental results on several benchmarks show that MarginGAN can provide improved accuracies and shorten training time as well.

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
