[Reviews · NeurIPS 2019]

Reviewer 1



1) What is the relationship of pseudo-ensemble to some classical ensemble methods, e.g., bagging or boosting? 2) For the semi-supervised experiment, the proposal needed to be compared with state-of-the-art methods. 3) It would be better to do experiments on more datasets because MNIST is a very simple dataset.

Reviewer 2



The originality of this paper is obvious, mainly due to the construction of the classifier. The paper is written well and has a significant contribution in the literature of SSL.

Reviewer 3



The main contribution of this paper is in setting up a 3 player game for semi-supervised learning where the generator tries to maximize the margin of the examples it generates in competition with a classifier + the traditional GAN approach of fooling a discriminator. This idea is novel to my knowledge. One small reservation I have with this method is that as the quality of the GAN and generated images increases the margin maximization for the classifier for generated examples becomes counter productive (as acknowledged by the authors) which requires careful early stopping. But this is standard practice with GANs and it should not be held against this paper. The paper is generally of high quality and significance but these could be improved by a broader treatment of related works. For instance, MixUp (Zhang et al, ICLR 2018) and CutMix methods (Yun et al, 2019) both generate artificial examples by combining real images with simple schemes to improve supervised learning performance. MixMatch (Berthelot et al, 2019) takes these ideas to the semi-supervised learning domain which forms an alternative to the GAN approach worthy of comparing to experimentally or at least in related work.

[Author Response · NeurIPS 2019]



Figure 1: Wrong pseudo labels will reduce the margins of examples.

## 1 Explanations

In the original version of the paper, we did not discuss why the proposed method works. Here we would like to give more explanations.

The reason why the accuracy improves is that the generated fake images can increase the classifier's tolerance to wrong pseudo labels, which is revealed in the ablation study of extreme training on MNIST. By training with the fake images, the classifier gets to know which kind of images are fake and don't belong to any real category. This will make the decision boundary shrink to surround the real images.

Figure 1 illustrates this intuition on a four-class problem. If the classifier chooses to believe the wrong pseudo labels, the decision boundaries have to stride over the "real" gap between the two classes of examples. But it will lead to reduced values in margin. So large margin classifiers should avoid to believing those wrong pseudo labels, and hence the accuracy can be improved.

## To Reviewer #1

We think our algorithm might be misunderstood as a pseudo ensemble method. We mentioned temporal emsembling and mean teach methods in the introduction section. But our work is to propose a new three-player game between the generator, the discriminator, and the classifier for semi-supervised learning. We can use mean teacher as our baseline model, but our method is essentially not a pseudo ensemble method.

For experimental comparisons, we first conduct preliminary experiment on MNIST and study the influence brought by the fake images. Then, we combine the proposed method with Mean Teacher and conduct experiments on SVHN, NORB and CIFAR-10. Our method is compared with state-of-the-art methods. The comparison results are shown at the top of page 7.

## To Reviewer #2

Question: It is known that GAN is not robust in general, and those SSL algorithms based on Psedudo labels may lead to an accumulation of error.

Answer: Our proposed method can greatly reduce the harm caused by the wrong pseudo labels, and please see the explanation at the top of this page. Of course GAN has its own weakness, but this is not our focus in this paper. We provided the source code and the same result can be obtained by directly running the code.

## To Reviewer #3

We will explore the relationship between our algorithm and prior work such as MixUp, CutMix, and MixMatch in the future version. And we should cite other recent works in SSL, like Sajjadi et al NIPS 2016 and Aila 2016. Also we will conduct experiments on STL10 to compare the competing methods. Thanks for these valuable comments.

[Meta-Review · NeurIPS 2019]

The paper formulates semi-supervised learning as a 3 player game among a generator, a classifier, and a discriminator. The generator and discriminator compete to train realistic examples, as in usual GANs, and the key new idea is that the classifier tries to maximize the margin of real examples and minimize the margin of fake examples. The method both improves predictive performance and greatly reduces training time. The reviewers agree that it is a significant contribution.